# The Effect of Annealing on Additive Manufactured ULTEM^™^ 9085 Mechanical Properties

**DOI:** 10.3390/ma14112907

**Published:** 2021-05-28

**Authors:** Yongjie Zhang, Seung Ki Moon

**Affiliations:** Singapore Centre for 3D Printing, School of Mechanical & Aerospace Engineering, Nanyang Technological University, 50 Nanyang Avenue, Singapore 639798, Singapore; ZH0007IE@e.ntu.edu.sg

**Keywords:** additive manufacturing, fused filament fabrication, thermal annealing, ULTEM™ 9085

## Abstract

Fused filament fabrication (FFF) is increasingly adopted for direct manufacturing of end use parts in an aviation industry. However, the application of FFF technique is still restricted to manufacturing low criticality lightly loaded parts, due to poor mechanical performance. To alleviate the mechanical performance issue, thermal annealing process is frequently utilized. However, problems such as distortion issues and the need for jigs and fixtures limit the effectiveness of the thermal annealing process, especially for low volume complex FFF parts. In this research, a novel low temperature thermal annealing is proposed to address the limitations in conventional annealing. A modified orthogonal array design is applied to investigate the performance of ULTEM™ 9085 FFF coupons. Further, the coupons are annealed with specialized support structures, which are co-printed with the coupons during the manufacturing process. Once the annealing process is completed, multiscale characterizations are performed to identify the mechanical properties of the specimens. Geometrical measurement of post annealed specimens indicates an expansion in the layering direction, which indicates relief of thermal stresses. Moreover, annealed coupons show an improvement in tensile strength and reduction in strain concentration. Mesostructure and fracture surface analysis indicate an increase in ductility and enhanced coalescence. This research shows that the proposed annealing methodology can be applied to enhance the mechanical performance of FFF parts without significant distortion.

## 1. Introduction

Additive Manufacturing (AM) is a layered manufacturing technique which is able to convert digital data into physical parts by deposition of feedstock through a layer by layer manner [1]. In an aviation industry, AM is gaining popularity, due to its potential of weight savings, reduced lead time and supply chain costs [2,3]. One of the most common methods is fused filament fabrication (FFF) as shown in Figure 1.

Although AM provides freedom in design and a host of aforementioned benefits, the material performance suffers due to the nature of manufacturing techniques [4]. Both static and dynamic mechanical performances are a fraction of the conventionally manufactured (e.g., injection molding) parts, and the part mechanical strength sensitive to the manufacturing process, such as locations, part build orientation, and printing parameters, etc. [5,6,7]. As a result, utilization of FFF parts is often restricted to manufacturing lightly loaded components in the aviation industry. These parts are not safety critical, and their mechanical performance is not a key design requirement.

To increase the utilization of FFF parts in more critical applications, there is a strong impetus to identify methods for improving the mechanical performance of the FFF parts. It is believed that thermal annealing holds the promise of alleviating the aforementioned issues, as the strength of the FFF parts depends highly on the melt bonding process of extruded filaments [5].

However, there are several drawbacks in using conventional thermal annealing process [5,8,9,10]. Firstly, if the annealing process is not well controlled, annealed parts may be severely distorted and original geometry may not be retained. Secondly, to retain the FFF part geometry, jigs and fixtures are a necessity, and utilizing such jigs and fixtures for highly complex FFF parts is not feasible. Lastly, AM enables small production lots, and manufacturing annealing jigs for low volume FFF parts can be extremely cost prohibitive.

Thus, in this paper, a novel methodology for annealing FFF parts is proposed to address the aforementioned limitations of conventional annealing process, while using the benefits of thermal annealing to improve the FFF parts mechanical performance to match the bulk material properties.

This paper is divided into following sections. Section 2 provides a literature review of the different methods that are used to increase the mechanical strength and theories on coalescence between adjacent rasters. Section 3 describes the derivation of the annealing temperature, and the various print and test setups. Section 4 discusses the outcome of the annealing process, which comprises of tensile testing, fractography and mesostructure analysis. The paper concludes with a summary of the contributions of this research and future work in Section 5.

## 2. Literature Review

It is well known that the strength of FFF parts depends strongly on how well the extruded material bonds or coalescences with adjacent material. To improve the bonding and the mechanical performance, it can be achieved via process parameter optimization, modifications to the printing process, or undergoing thermal process post printing.

There are many literature focusing on predicting the mechanical performance and maximizing the achievable strength of FFF parts using process parameter optimization. Rodriguez et al. [11] investigated the void and bond length density between adjacent rasters, based on process parameters, namely the air gap and extrusion flow rate. Coogan and Kazmer [12] developed a healing model for predicting the bond strength of FFF parts, which depended on bonding strength estimated from wetting and reptation theory. Shelton et al. [13] investigated the impact of thermal process, in particular, build chamber temperature, on the interlayer strength for ULTEM™ 9085 parts. It was found that higher the chamber temperature, the more consistent was the interlayer necking, which resulted in higher mechanical properties. Kumar et al. [14] utilized pore size and neck length to predict the mechanical properties of FFF parts and provided relatively good predictive accuracy.

Garg and Bhattacharya [15] developed a modelling technique to identify necking (both inter layer and intra layer) and air voids. They also explored the effect of the layer thickness and different raster angle on the mechanical properties. Ahn et al. [16] investigated various print parameters such as air gap, road width, model temperature and orientation of rasters in a design of experiment and compared the mechanical properties against an injection molded specimen. They derived various build rules, such as recommending negative airgap to increase strength and stiffness and aligning the road in the loading direction. Zaldivar et al. [17] studied the part build orientation effect on the mechanical properties and compared it to conventionally injection molded ULTEM™ 9085 specimens. It was found that the highest achievable strength (in terms of orientation) for FFF was 86% of the injection molded parts. The weakest orientation could go as low as 52%. Forés-Garriga et al. [18] performed a comprehensive study on the role of infill in ULTEM™ 9085 on mechanical performance, and weight reduction. Specimens with different infill parameters and orientation were printed and characterized, and the compliance matrixes were established for the infill parameters. Singh et al. [19] studied the feasibility of FFF multi-material printing to provide FFF parts with enhanced multi-dimensional properties.

On the other hand, there are several literatures that focuses on improving the mechanical properties of FFF parts without parameter optimization. Some literature focuses on innovative methods such as process modification of FFF parts to incorporate additional energy sources with improving bonding. Luo et al. [20] included an additional laser source to improve the interlayer shear strength of FFF parts. Nycz et al. [21] utilized infrared heating to improve interlayer bonding. Other literature investigates thermal process post printing for properties enhancement. Sardinha et al. [22] explored the impact of pass the top surface of the part with the nozzle to improve surface roughness and interlayer strength. Parker et al. [23] described a post processing technique using hot isostatic press various thermoplastic, including ULTEM™ 9085. Porosity of the FFF printed coupons reduced but the authors did not find a significant trend in the strengthening of coupons. Wang et al. [24] proposed a novel approach of including thermally expandable microspheres, which expanded 50–100 times the volume when heated above a threshold temperature. These additives increased the mechanic properties compressive strength increased 3-fold and reduced the porosity of the printed parts. However, these literatures demonstrated that current heat annealing methods required molds and jigs to minimize specimen warpage [5,8]. The usage of these molds will diminish the usefulness of the 3D printed parts, as complex 3D printed parts are not able to successfully anneal without additional jigs, and hence, greatly reducing the effectiveness of additively manufactured parts [5].

To understand the factor driving the bond formation and coalescences, several theories have been proposed. During the FFF process, the depositing material, which is at temperature higher than its glass transition temperature (*T_g_*), is brought into contact with the existing extruded material, and the polymer molecules are allowed to inter-diffuse and form bonds, which is similar to polymer welding. The bond strength is driven by factors such as time, contact area between the rasters, and the thermal history of the interfaces, i.e., heating up from build chamber temperature when a new material is deposited [8,25]. N. Turner, Strong, and A. Gold [25] described the different stages of the polymer bond formation, namely surface contact, neck formation, diffusion and neck growth, and randomization. Reptation theory of molecular diffusion, developed by de Gennes [26], was used to explain the bond formation for FFF process in several papers [5,8].

On the other hand, Luchinsky, Hafiychuk, Hafiychuk, and Wheeler [9] found that the reptation theory was insufficient in modelling the annealing time required, and the empirical time required was many magnitudes higher than what was predicted by the reputation theory. Moreover, the rigidity and charge of the polymer chains were not taken into consideration. Prajapati et al. [27] proposed Arrhenius kinetics theory to model the thermal annealing of FFF parts. The theory relates the neck size growth with the exponent of activation energy, universal gas constant and the temperature at which the annealing is performed. Malekmotiei et al. [28] discussed 3 different temperatures the plastic that was annealed—below, at, and above *T_g_*. It was concluded that annealing at temperatures lower than *T_g_* was beneficial as it enhanced the mechanical properties. Annealing around *T_g_* decreases yield stress and Young’s Modulus, and that annealing at temperature higher than *T_g_* can cause the polymer to become more brittle.

## 3. Experiment Setup

Most of the literature focuses on the readily available thermoplastic, such as ABS and Poly-Latic-Acid (PLA) which are certainly and readily printable on a desktop printer, while there are limited literatures on engineering thermoplastic. ULTEM™ 9085 is a special mix of Poly-Carbonate (PC) and polyetherimide (PEI), which provides the thermoplastic with the strength and toughness required for many applications, in particular for aerospace applications. It is able to retain good mechanical properties at elevated temperatures up to 153 °C and is able to meet flame, smoke, and toxicity requirements, which are critical to an aviation industry [29,30,31,32,33]. For example, Airbus has used more than 1000 ULTEM™ 9085 FFF cabin interior parts in the A350XWB [32]. Although ULTEM™ 9085 has been recognized as suitable for aerospace applications, however, it is not widely studied and characterized, due to the high material cost and the difficulty in printing the material, which requires a costly setup (i.e., requires high build chamber temperature and high deposition nozzle temperature).

### 3.1. Derivation of Annleaing Specifications

The first batch of coupon was annealed using a preliminary annealing cycle to observe the effect of annealing on ULTEM™ 9085. A holding temperature of 204 °C was selected and it was found that the coupons warped and deformed. Similar deformation was seen in Hart, Dunn, Sietins, Mock, Mackay, and Wetzel [5], where the FFF coupons were directly annealed without any supporting fixtures. Hence, it is validated that prolonged exposure to temperatures exceeding the *T_g_* of ULTEM™ 9085 (170 °C–178 °C, depending on the moisture content [10]) causes the part to deform uncontrollably.

In the proposed annealing specification, the holding temperature was lowered. In addition, support structures were modified to serve as a fixture for stabilizing the dimensions of the coupons during the annealing process. Due to the higher extrusion temperature of the support material, it is postulated that it is able to retain its geometry better than ULTEM™ 9085 when subjected to heat. The following section provides the derivation of the annealing temperature.

Blanco, Cicala, Ognibene, Rapisarda, and Recca [31] explored several blends of Poly-Carbonate and Polyetherimide and compared against ULTEM™ 9085. Through analyzing the degradation behavior of the different blends using DTG (Derivative thermogravimetry) and DSC (Differential Scanning Calorimetry) curves, it was found that the PC and PEI were an immiscible blend, with some regions of partial miscibility. It is further proposed that PC was added to PEI to reduce the viscosity, which aided the extrusion and processability of the polymer blend. It has been found that ULTEM™ 9085 exhibited properties very close to blends of 80% PEI and 20% PC [9,30]. As a result, there are distinct peaks of *T_g_* at 212 °C and 141 °C which may belong to PEI and PC, respectively [30,34]. Dawkins et al. [35] showed that immiscible blends of polymer could be phase separated once heated above the *T_g_* and they did not remix upon cooling. As such, it is vital that the annealing temperature does not exceed *T_g_* significantly to reduce the potential separation.

To derive the annealing time, the following Equation (1) is utilized.
(1)$$t_{s}=\mu r/\Gamma$$
where $t_{s}$ is the characteristic “sintering” time scale, μ is the characteristic viscosity, r is the characteristics radius of the free surface driving coalescence, and Γ is the surface tension [5]. Typical values for polyetherimides for the surface tension ranges from 38 to 50 dyn/cm, zero shear viscosity is 2000–5000 Pa·s [9], which are summarized in Table 1. Using these values for estimating the range of sintering time scale, it ranges from 56 h to 186 h.

A designed experiment using a modified L9 orthogonal array based on Taguchi et al. [36] is applied to investigate the effect of annealing time, temperature and print direction, as shown in Table 2. Levels for temperature factor are selected to be 170 °C and 180 °C. 170 °C is selected to be the lower level, a temperature lower than *T_g_*. The higher level, 180 °C is selected to investigate the effect of the holding temperature beyond *T_g_*. As for the duration of the annealing process, 24 h and 96 h are selected, which are sufficiently far apart to investigate the effect of sintering time on the mechanical properties. Due to the long annealing duration, mixed level design is used to minimize the overall experimental time by reducing the time and temperature factors to 2 levels (see Table 2). Table 3 shows the different run configurations, and the run number corresponds with the coupon number.

### 3.2. Coupons and Orientation in Experiment

Standardized ISO 527-2 Type 1A tensile coupon geometry was selected for this investigation. Figure 2 shows the geometry of the coupon and Table 4 shows the nominal dimensions.

### 3.3. Printing Parameters

The coupons are prepared using CAD program, which are subsequently sliced and converted into a print file. Table 5 shows the printing parameters to fabricate the coupons. The used nomenclature can be found in Figure 1. Coupons in the second round are printed using the surround support option to provide support and stability during the annealing process.

### 3.4. Direct Image Correlation

In this paper, 2D Digital Image Correlation (DIC) is utilized to investigate the full strain field of the coupons undergoing tensile testing. DIC is an optical method to measure the evolving full-field surface deformation and shape of test article through a mechanical test. Stochastic patterns are utilized to describe image areas, which can be determined with sub pixel accuracy by analyzing the image information. The stochastic patterns are applied manually to the surface of the specimens with a light coat of white paint followed by speckles of black paint. During the post processing phase, digital grids are superimposed onto captured images. These temporary successive captured images are then analyzed to calculate derived field, such as displacement and strain fields. As it is a non-contact method of measurement, it is significantly versatile and can be used in wide ranging of different applications [37,38]. In this investigation, sequential images of the tensile tests are captured at a rate of 25 Hz. Subsequently, the force readings from the tensile test machine are then combined with these strain field readings. The images are then processed using GOM Correlate software, using bicubic subpixel interpolation with a facet size of 15 pixels and point distance of 10 pixels. Figure 3 shows the setup of the DIC. The long axis of the camera is affixed parallelly to the long axis of the test piece and to ensure that the stereo plane is perpendicular to the test piece.

### 3.5. Equipment

The coupons were printed on a Fortus^®^ 450MC and were thermally annealed in EYELA NDO-451SD natural convention oven with automated program control. Ramp up time was approximately 30 min to reach the defined temperature, and the cool down duration to room temperature was set to 45 min to ensure that the coupons were slowly brought to room temperature without inducing further thermal stress. Figure 4 provides a graphical view of the temperature profile for the annealing process. The tensile tests were on a 10kN Shimadzu Tensile Testing machine. Fractography pictures were captured using JOEL Scanning Electron Microscope (JSM-5600LV). Gold spluttering was performed on the fracture surface prior to examining the specimens under the SEM. Pictures of mesostructure was captured using optical microscopy with an Olympus BX53M fitted with a DP22 lens, using 50× magnification. The mesostructure specimens were prepared by sanding with a 180-grit sandpaper followed by polishing with a 1 µm diamond paste.

## 4. Experimental Results

This section discusses the various methodologies to characterize the annealed coupons and analyze the results. Characterization includes geometrical measurements to evaluate the dimensional stability of coupons. Tensile test is performed to understand the improvement in mechanical properties. Finally, mesostructure and fractography analysis is performed to observe any change in mesostructure and correlate any improvement in the tensile test. Coupon 8 was deformed during the thermal annealing, and hence, characterization was not performed.

### 4.1. Coupon Dimensional Measurement

Measurement of the dimensions was conducted to determine the effect of annealing on the dimensional accuracy. The measured dimensions of the thermal annealed specimen were normalized against the average dimensions of the printed specimen using the following Equation (2).
(2)$$\Delta L=\frac{L_{f}}{L_{i}}\times 100$$
where $L_{f}$ is the measured dimension after annealing, $L_{i}$ is the measured dimensions prior to annealing, and ΔL is the dimensional change in various directions.

Results from the dimensional analysis are presented in Table 6, and the nomenclature of the measurement is described in Table 4. The print direction is shown in Figure 5a–c and photographs of annealed coupons are shown in Figure 5d–f. Further, the average dimensional changes of the annealed coupons in each of the axis (∆X, ∆Y, ∆Z) are also presented. Most of the dimensional changes are well with 5%, which signifies the dimensional stability of the proposed annealing process.

One major observed trend is that coupons are expanded in the layering direction (i.e., in Z axis), and the orthogonal dimensions shrunk, which is to retain the same total volume. For coupons 1, 5, and 9, the thickness of the coupon increases, as evident by the increase in *h* (gauge) and *h* (grip area). For coupons 2, 6, and 7, the width of the coupons increases, as shown by the positive increment in the $b_{2}$ and $b_{1}$ measurement. For coupons 3 and 4, the length of the coupons increased marginally, but exhibited the same trend as the coupons printed in the XZ direction (i.e., expanded in the width).

To identify the main effect from the dimensional changes, analysis of the modified L9 design is conducted in Minitab^®^ 19. Figure 6 shows the effect plot for the dimensional change in the X, Y, and Z direction, respectively.

In general, the print direction of the coupons has the most significant impact on the change in dimensional. From Figure 6a, the coupons shrunk in length in the X direction, by an average of 1.05%. It is observed that the higher the annealing temperature, and longer the annealing process, the more the coupons shrunk length in X direction. Coupons printed in the XY had the largest shrinkage, by 1.77% in the X direction. From in Figure 6b, the annealed coupons expanded by an average of 0.06% in the Y direction. ZX coupons expanded by an average of 3.05%, whereas XY and XZ contracted approximately 2%. In the Z direction (i.e., the layering direction, as shown in Figure 6c, the coupons expanded by an average of 2.22%. Print direction has the largest effect on the dimensional change in the Z direction, followed by temperature and duration. XY coupons expanded by an average of 5.07%, whereas XZ and ZX expanded approximately 1.57% and 0.35%, respectively. Coupons experience larger change in dimensions when they are thermally annealed at a higher temperature and for a longer time. For example, Coupons 5 and 9 tend to have larger magnitude of dimensional change as compared to Coupon 1, which underwent a milder thermal annealing process.

The uniform trend of expansion in the Z direction can be explained by the reduction of the residual thermal stress in the printed coupons. After a layer is deposited, the material, cools down from the extrusion temperature, to approximately the chamber temperature. When the next layer is deposited, the temperature of the previous layers instantaneously rises, to near the extrusion temperature [31], resulting in a large and sudden expansion of the previous layer, which introduces these in-built thermal stresses. Through thermal annealing, it is believed that these residual stresses are alleviated, which resulted in an expansion in the layering.

The apparent densities of the coupons were determined to identify whether the specimens became denser, with lesser voids. A volume displacement method was utilized to for the characterization, by immersing the sample into a known volume of liquid and determining the resultant volume change. To derive the apparent density, the following Equation (3) was used,
(3)$$D_{app}=\frac{M}{V_{app}}$$
where M is the mass measured from weighing scales, and $V_{app}$ is the apparent volume measured using the volume displacement method.

The average apparent density of the samples that underwent thermal annealing was found to be around 1.24 g/cm^3^ with a coefficient of variance of 3.0%. The average apparent density of the printed samples was slightly lower, at around 1.22 g/cm^3^, with a higher coefficient of variance of 3.2%. This suggests that there is consolidation and coalescing of rasters, resulting in slightly higher densities.

### 4.2. Mechanical Properties

Results are tabulated in Table 7 and Table 8, which shows the mechanical properties for control coupons and heat annealed coupons, respectively. For Coupons 3 and 4, the results were much lower than the expected, both in terms of the ultimate tensile strength and the strain at break. Upon further investigation, it was found that there were defects embedded in between layers of the 2 specimens, which might be caused by deposition of stray filaments. From Table 8, it is clear that although the coupons can reach close bulk material strength (up to 89% of the IM strength), the strain at break is much lower as compared to the IM specimen, often falling below 10% of the reported IM values. Annealed XY coupons have significantly higher strain at break, up to 27% improvement as compared to the control coupons. However, an approximately 3.4% drop in ultimate tensile strength is observed. Annealed XZ coupons, on the other hand, shows improvement in both ultimate tensile strength and higher strain at break, albeit the increase is lower, at 1.4% and 5.7%, respectively.

Figure 7 compares the stress strain curve of the annealed coupons and representative control coupons. From Figure 7, it is evident that there is distinct clustering of the stress strain curves according to the different print directions. Further, there are indications of the annealed coupons having higher ductility and higher tensile strength. Additionally, Figure 8 and Figure 9 explore the differences in details.

Figure 8 compares annealed coupons and the representative control coupon printed in the XY direction. From Figure 8, it is evident that the annealed coupons reach the plateau indicating plastic deformation sooner, while exhibiting higher strain at break and ductility as compared to the unannealed samples. However, higher ductility is traded off for a slight decrease in tensile strength, as compared to the control. Coupon 1 is an anomaly, as both strength and strain at break fall short of the unannealed sample.

Figure 9 compares annealed coupons and the representative control coupon printed in the XZ direction. From Figure 9, it is evident that the annealed coupons tend to have closer spread for both the tensile strength and the elongation at break of the coupons. The highest tensile strength was achieved by the thermally annealed coupon, reaching a strength up to 89% of the IM specimen.

To identify the significance of the effect on annealing on both ultimate tensile strength and strain at break, the effects plot is plotted as shown in Figure 10a,b, respectively. In both cases, the variation in tensile properties due to the change in print direction is much more significant than the effect of annealing in this experiment. This means that the annealing did not achieve a significant reduction in anisotropy resulting from the different print direction. Nonetheless, it is observed that longer and higher annealing temperature improved mechanical properties.

### 4.3. Fractography Anaylsis

Scanning Electron Microscopy was performed on both annealed and control coupons. Both ductile and brittle failure regions are observed on the fracture surface on all coupons. It is postulated that fracture begins with a brittle morphology, characterized by a relatively flat surface with little crazing. As the crack propagated through the cross sectional area, it changes to a more ductile morphology, with an increase in the stress whitening or crazing of the individual filament. Such response to stress is often seen in polymers exhibiting fibrillar characteristics [40].

For Figure 11a–d, the samples were printed in the XY direction. Although distinct layers are still observed for thermally annealed specimens, it is observed that fracture propagates though the different layers, which signifies the improved interlayer coalescence, in contrast with control coupon (Figure 11d). Higher ductility, as evident from the increased necking of the rasters and significantly more distinct chervons and striation marks, is observed in Coupons 1, 9, 5. This accounts for the increased strain at break in the tensile test results.

For Figure 12a–d, the samples were printed in the XZ direction. In general, higher degree of coalescence is observed as compared to the XY specimens, which can account for the much higher average tensile strength of the XZ coupons. Comparison between Figure 12a,b (Coupons 2 and 6) indicates that there is no significant difference in mesostructure when there is an increase in the annealing time and temperature. Coupon 7 and Coupon XZ2 indicate similar tear out failure mode, whereby the fracture propagates through the length of the raster. This signifies that as during the printing of XZ specimens, the degree of coalescence between the rasters are sufficiently high and that thermal annealing has limited further improve the coalescence.

For Figure 13a–c, the samples were printed in the ZX direction. From the fracture surfaces, it is apparent that the specimens failed between the layers due to weak interlayer adhesion. From Figure 13a,b, the imprints from the layer above can be seen. This signifies that there is softening of the ULTEM ^TM^ 9085 material, and possibly some degree of coalescence that has occurred. This is in contrast with Figure 13c (Coupon ZX2). Further, the imprint lines are significantly more distinct on Coupon 3 (Figure 13a) as compared to Coupon 4 (Figure 13b), which could be due to the effect of higher annealing temperature experienced by Coupon 3. Although it appears that the annealing improved the interlayer bonding, however, due to printing defects introduced in Coupons 3 and 4, the tensile strength and the strain at break is lower as that the control coupon.

In summary, across the different print orientation, it is apparent that there is an increased degree of coalescence after the annealing is performed on the ULTEM™ 9085 parts. Increased ductility is observed in the annealed specimens printed in XY and XZ. For the ZX specimens, there is an increase in interlayer coalescence. However, due to the printing defects, the tensile strength is not improved.

### 4.4. Mesostructure Analysis

Mesostructure analysis is performed on selected specimens. For ZX specimens, the results are not discussed as appearance of voids depends heavily on the slicing plane. Figure 14a,b show the annealed specimens and as printed specimens for XY direction, and Figure 14c,d show the annealed specimens and as printed specimens for XZ direction. The annealed specimens have a significantly lesser void content in the bulk mesostructure (Figure 14a,c), which signifies that annealing improves coalescence between adjacent rasters that decreases the voids.

### 4.5. DIC Results

Figure 15a,b show the annealed and control coupons for the XY orientation, and Figure 15c,d show the annealed and control coupons for the XZ orientation. The DIC images depicts the coupons close to the breaking point. From the images, it is apparent that the strain distribution on the surface for the annealed specimen is more homogenous. This is in contrast with Figure 15b, where the crisscross pattern is distinct throughout the gage length of the specimen, which closely resembles the pattern of the rasters.

Similarly, the annealed specimen in Figure 15c shows more homogenous strain distributions in the gage length, whereas the unannealed specimen Figure 15d shows distinct regions of stress concentration. As such, it is evident that the annealing process alleviates the strain and stress concentration when the specimens are subjected to loading, which delays the onset of the failure of the specimens.

## 5. Conclusions

In this paper, a novel thermal annealing methodology was proposed to improve the mechanical properties of FFF parts. Using the proposed annealing methodology, it is shown that limitations with conventional thermal annealing, such as significant distortions and the need for jigs can fixtures, can be alleviated. Further, from the multiscale characterization performed in this research, the following is observed:All annealed coupons have expanded in the layering direction, indicating a thermal stress relief. This signifies that the FFF prints are significantly stressed in the direction of layering, which will subsequently expand once it has been annealed close to the *T_g_* for prolonged periods of time.Coupons are generally more ductile after annealing, achieving up to 27% and 5% improvement for XY and XZ annealed coupons, respectively. DIC results show a reduction in strain concentration, which decreases the likelihood of localized failure due to high strain concentration. Tensile strength, on the other hand, dropped slightly about 3.4% and increased 1.4% for XY and XZ annealed specimens, respectively.Mesostructure and fracture surface analysis shows increased coalescence and chevrons indicating higher degree of necking during fracture, a sign of increased ductility.It was shown that thermal annealing can be carried out without the aid of specialized molds, with most of the coupons are still dimensionally stable, retaining its geometry.

Limitations of this research includes exploration of only a single annealing profile, which can limit the effectiveness of the thermal annealing process. Future work can include study of the annealing cycles. Exploration of long-term fatigue performance for these annealed parts is also another key research area as it has been shown in this research that annealing can relieve of thermal stresses in FFF parts, which can aid in fatigue performance.

## Figures and Tables

**Figure 1 materials-14-02907-f001:**
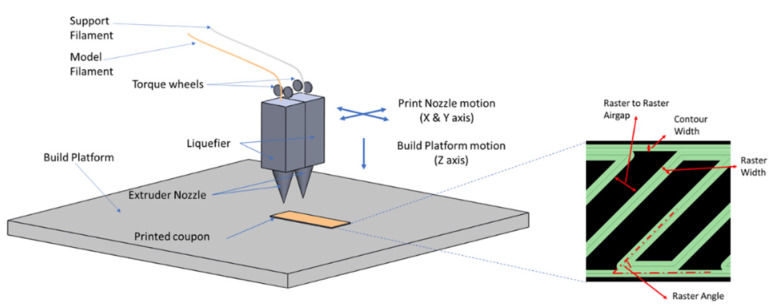
Principle of fused filament fabrication.

**Figure 2 materials-14-02907-f002:**
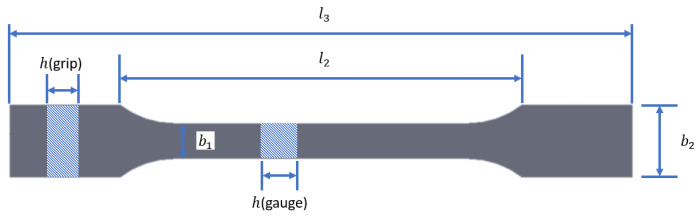
ISO 527-2 Type 1A tensile coupon and nomenclature.

**Figure 3 materials-14-02907-f003:**
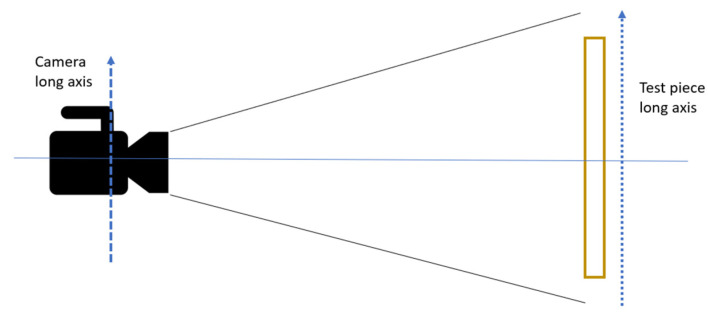
Digital Image Correlation setup used in measurement of the tensile properties.

**Figure 4 materials-14-02907-f004:**
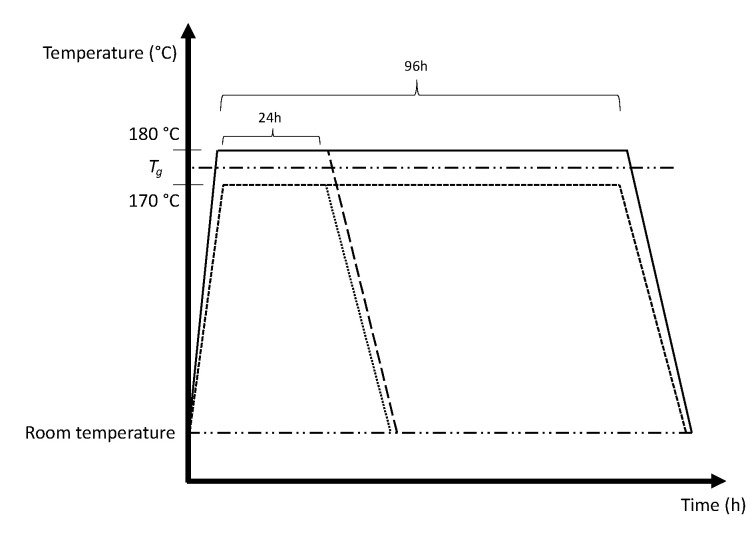
Temperature profiles for the annealing process.

**Figure 5 materials-14-02907-f005:**
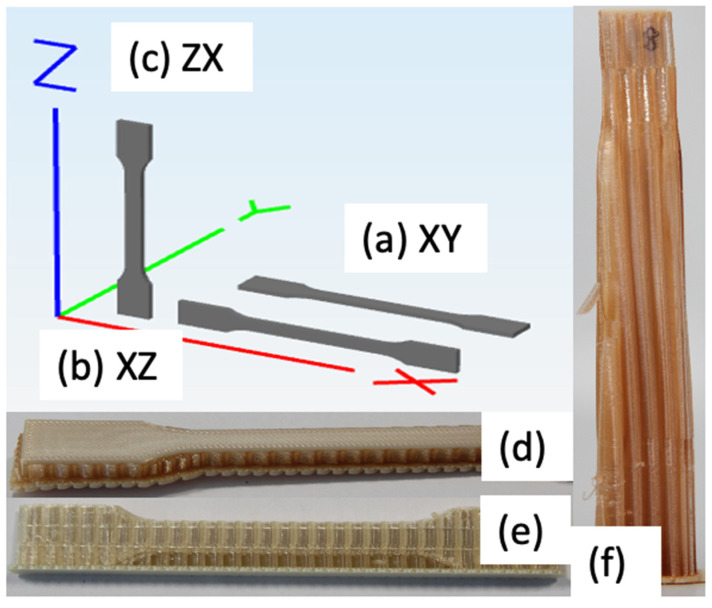
X, Y, and Z axis for coupons: (**a**) XY (Coupons 1, 5, 9), (**b**) XZ (Coupons 2, 6, 7), and (**c**) ZX (Coupons 3, 4, 8). Annealed specimens with co-printed supporting structures (**d**) Coupon 1, (**e**) Coupon 6, (**f**) Coupon 8.

**Figure 6 materials-14-02907-f006:**
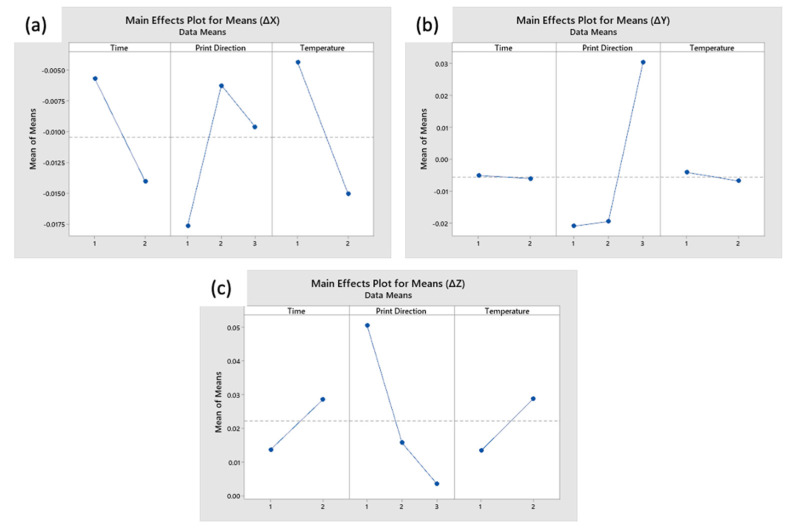
Effect plot for change in dimensions in the (**a**) X direction, (**b**) Y direction, and (**c**) Z direction.

**Figure 7 materials-14-02907-f007:**
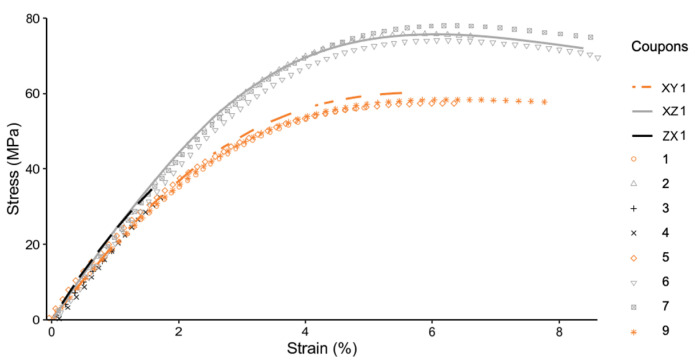
Stress–Strain curve for all coupons (excluding the injection molded).

**Figure 8 materials-14-02907-f008:**
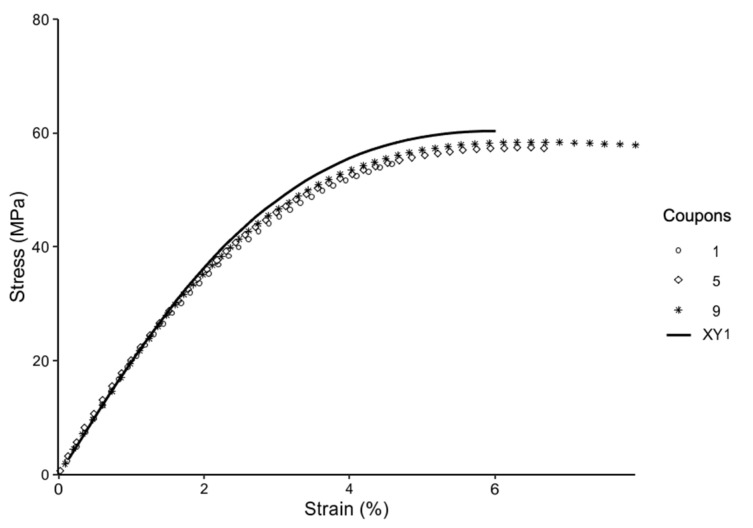
Stress–Strain curve for coupons printed in XY, annealed (1,5,9), and control (XY1).

**Figure 9 materials-14-02907-f009:**
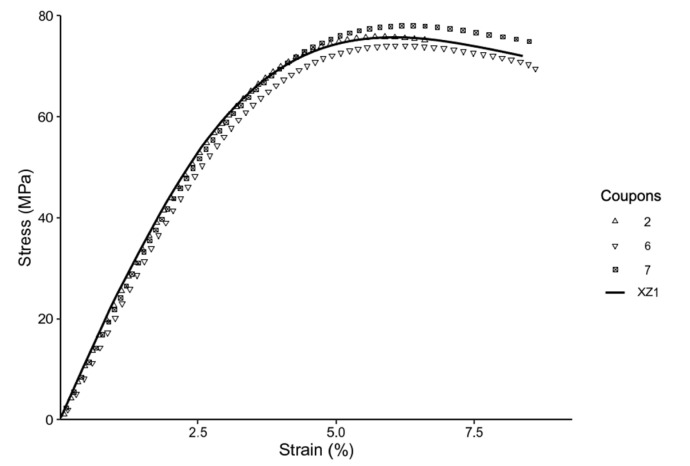
Stress–Strain curve for coupons printed in XZ, annealed (2,6,7), and controls (XZ1).

**Figure 10 materials-14-02907-f010:**
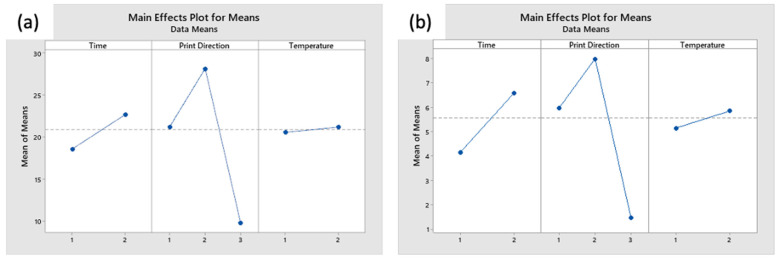
Effect plot for change post annealing: (**a**) Ultimate Tensile Strength and (**b**) Strain at Break.

**Figure 11 materials-14-02907-f011:**
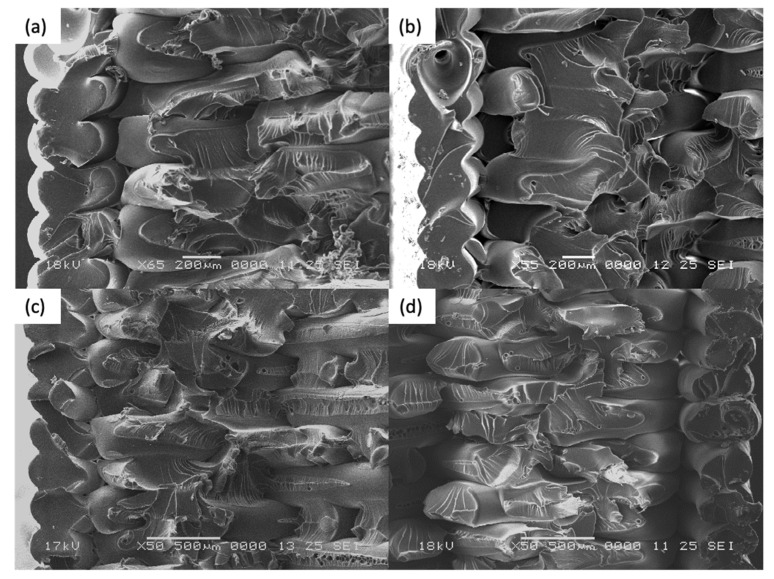
Fracture surface for (**a**) Coupon 1, (**b**) Coupon 9, (**c**) Coupon 5, and (**d**) Coupon XY1.

**Figure 12 materials-14-02907-f012:**
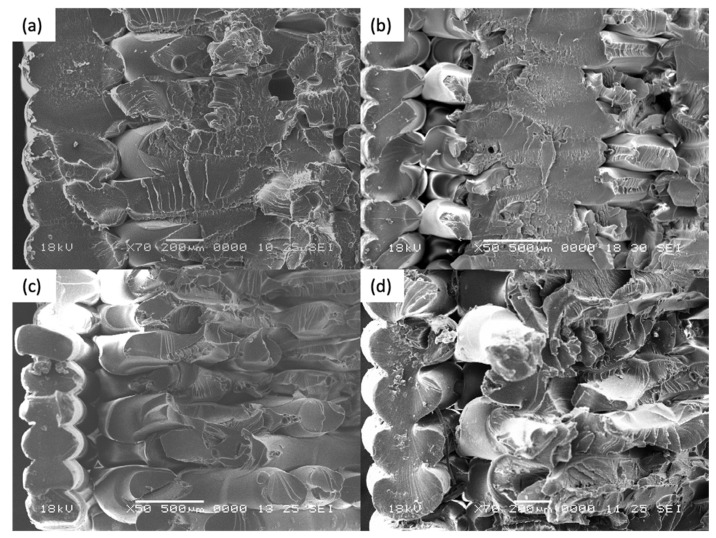
Fracture surface for (**a**) Coupon 2, (**b**) Coupon 6, (**c**) Coupon 7, and (**d**) Coupon XZ2.

**Figure 13 materials-14-02907-f013:**
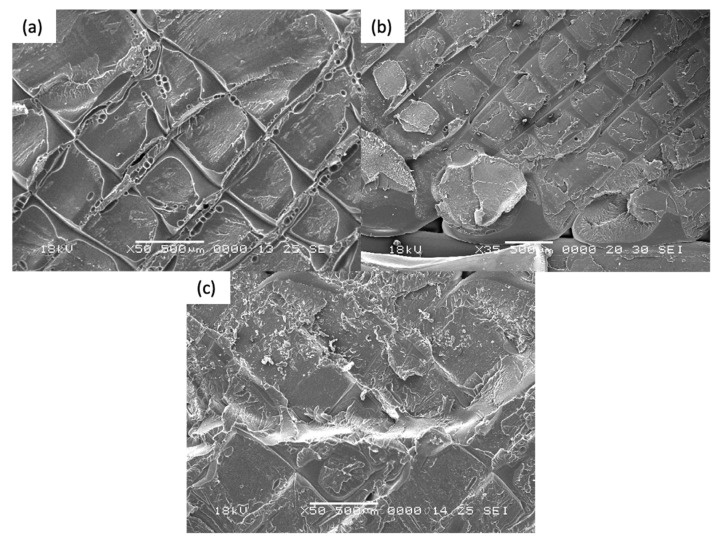
Fracture surface for (**a**) Coupon 3, (**b**) Coupon 4, and (**c**) Coupon ZX2.

**Figure 14 materials-14-02907-f014:**
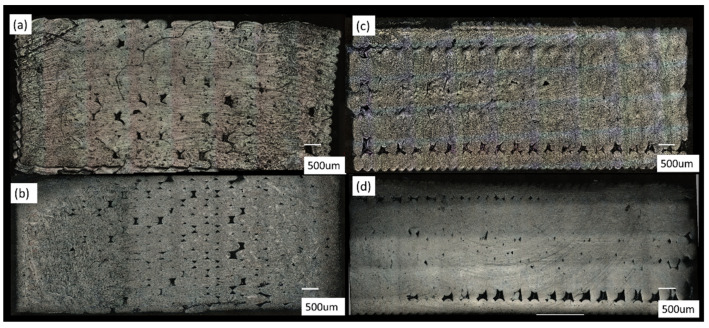
Mesostructure for (**a**) Coupon 9 , (**b**) Coupon XY2 , (**c)** Coupon 7 , and (**d**) Coupon ZX1.

**Figure 15 materials-14-02907-f015:**
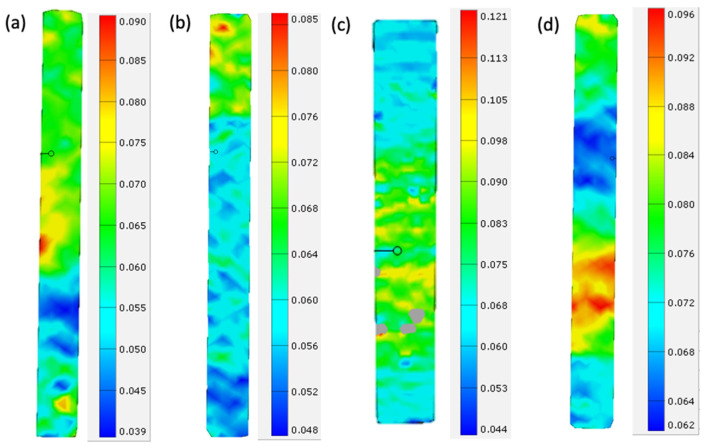
DIC images for Specimen (**a**) 5, ε = 6.2%, (**b**) XY2, ε = 6.0%, (**c**) 7, ε = 7.6%, and (**d**) XZ2, ε = 7.6%.

**Table 1 materials-14-02907-t001:** Parameters Γ, μ,r
to estimate $t_{s}$.

Description	Values
Surface tension, Γ	38–50 × 10^−7^ N/m
Zero shear viscosity/Characteristics viscosity, μ	2000–5000 Pa · s
Characteristics radius, r	0.508 mm

**Table 2 materials-14-02907-t002:** Factors used in modified L9 orthogonal array.

Factor	Level 1	Level 2	Level 3
Time	24 h	96 h	-
Print Direction	XY	XZ	ZX
Temperature	170 °C	180 °C	-

**Table 3 materials-14-02907-t003:** Modified L9 Orthogonal Array used in this experiment, with the respective factor levels from Table 2.

Run/Coupon	Time (h)	Print Direction	Temperature (°C)
1	24	XY	170
2	24	XZ	180
3	24	ZX	180
4	96	ZX	170
5	96	XY	180
6	96	XZ	180
7	96	XZ	170
8	96	ZX	180
9	96	XY	180

**Table 4 materials-14-02907-t004:** Dimensions to be measured and their nominal value.

Dimensions	Values (mm)
$l_{3}$	170
$b_{2}$	20
$b_{1}$	10.0
h (gauge)	4.0
h (grip)	4.0

**Table 5 materials-14-02907-t005:** Printing Parameters.

Description of Parameters	Values
Raster Width	0.508 mm
Contour Width	0.508 mm
Slice Height	0.254 mm
Contour to Raster Air Gap	0 mm
Raster to Raster Air Gap	0 mm
Raster angles	+45°/−45°
Layers between alternating of raster angles	1

**Table 6 materials-14-02907-t006:** Dimensional variations of coupons after thermal annealing.

**Coupon**	$l_{3}$	$b_{2}$	$b_{1}$	h (gauge)	h (grip)	∆X	∆Y	∆Z
1 (XY)	−0.49%	−0.43%	0.17%	2.12%	3.05%	−0.49%	−0.13%	2.59%
5 (XY)	−2.84%	−3.88%	−4.03%	6.72%	6.83%	−2.84%	−3.96%	6.78%
9 (XY)	−3.24%	−4.26%	−4.03%	7.76%	8.92%	−3.24%	−4.15%	8.34%
2 (XZ)	0.00%	0.82%	1.64%	−6.61%	−1.38%	0.00%	−4.00%	1.23%
6 (XZ)	−1.77%	2.07%	2.82%	−6.30%	11.77%	−1.77%	2.74%	2.45%
7 (XZ)	−0.10%	0.74%	1.34%	−7.00%	−2.19%	−0.10%	−4.60%	1.04%
3 (ZX)	0.30%	−1.21%	−2.16%	2.18%	3.02%	−1.21%	2.60%	0.30%
4 (ZX)	0.39%	−0.71%	−1.32%	2.91%	4.09%	−0.71%	3.50%	0.39%

**Table 7 materials-14-02907-t007:** Mechanical properties for control specimens in ZX, XY, and XZ.

	Ultimate Tensile Strength (MPa)	% of IM	Strain at Break (%)
Injection Molded [39]	88.00	-	72.0
ZX1	37.21	42%	2.01
ZX2	38.38	44%	2.00
ZX3	39.40	45%	2.21
XY1	60.38	69%	5.90
XY2	61.47	70%	6.16
XY3	59.63	68%	5.60
XZ1	75.87	86%	8.44
XZ2	75.48	86%	9.30
XZ3	74.49	85%	7.73

**Table 8 materials-14-02907-t008:** Mechanical properties for annealed specimens.

Coupon	Time	Print Direction	Temperature	Ultimate Tensile Strength (MPa)	% of IM	Strain at Break (%)
Injection Molded [39]	-	-	-	88.00	-	72.0
1	24	XY	170	56.48	64%	4.99
2	24	XZ	180	75.87	86%	6.38
3	24	ZX	180	21.10	24%	1.09
4	96	ZX	170	33.91	39%	1.83
5	96	XY	180	57.49	65%	6.38
6	96	XZ	180	74.18	84%	8.97
7	96	XZ	170	78.11	89%	8.63
8	96	ZX	180	-	-	-
9	96	XY	180	58.46	66%	7.53

## Data Availability

The data are not publicly available due to commercial confidentiality.

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
