# Peer review of "The Effect of Annealing on Additive Manufactured ULTEMâ„¢ 9085 Mechanical Properties"

_materials, 2021, doi:10.3390/ma14112907_

Round 1

Reviewer 1 Report

The paper presents results of experimental investigations of effect of annealing on additive manufactured ULTEM™ 9085 mechanical properties. The paper presents well planned and performed experimental work, providing new input to the knowledge on additive manufacturing technologies, which still require more understanding of materials and processes. However, there are several critical aspects of the work to be pointed out:

  1. Fused Deposition Modeling (FDM), which is a trademark from Stratasys, for the standard terminology for the additive manufacturing process may be consult ISO/ASTM 52900:2015(E), maybe something like: material extrusion additive manufacturing or fused filament fabrication (FFF).
  2. The abstract is very general, and there is a lot of irrelevant information. The abstract should be explained and showed the important aspects of work. So, this abstract in the present form is unacceptable.
  3. There is no engineering context within which the research is placed. What is the intended use of this filament? What parts can be made with this type of filament? Where can they be used in aviation?
  4. As the data analysis of this article, more statistical analysis should been added, for example, the coefficient of variation (CV, standard deviation) of the experimental data should been added and discussed. This lack of statistics is not acceptable in professional publications and needs to be completed.
  5. There was a problem with the calibration of the tensile machines. Why does the curve of 7 (figure 9) start at significantly higher stress compared to the other variants. Please clarify.
  6. The quality of the figures must be substantially improved. The curves fron Figure 7 are difficult to identify.
  7. There are no images with the specimens manufactured by material extrusion 3D printing process.

Author Response

Refer to the attached rebuttal letter.

Reviewer 2 Report

General comments:

In this work the authors study the effect of annealing ULTEM 9085 printed parts, at a temperature close to the material glass transition temperature, on their mechanical performance and dimensional stability.

Overall, the English is very poor, turning difficult or impossible, sometimes, the interpretation of the text.

Having in mind the objective of the present study, it would be relevant to refer, in the state-of-the-art, alternative strategies already used by other researchers to improve the mechanical performance of parts printed by this technique. Some examples are the use of a laser during the printing stage, the use of heaters supported by the extrusion head (that heat the previously deposited raster in the location where the new will be deposited), among other. Also, the results are the expected ones.

Other questions:

  • In Figure 5, what is the difference between (a) and (c) printing conditions? What is the rationale behind the nomenclature adopted for the printing orientation? I cannot understand why (a) is XY and etc. Probably it would be better to call them ‘printed over the length and width (a)’, ‘over the length and thickness (b)’ and ‘over the width and thickness (c)’...
  • In Figure 14, (d) seems to have bigger voids than (c), i.e., despite of a better coalescence, it seems that the porosity of the sample increased ... Is it true?
  • The caption of figure 17 is not clear and is incomplete, since (c) and (d) are not defined.

Author Response

Refer to the attached rebuttal letter.

Reviewer 3 Report

The paper discusses the mechanical properties of FDM Ultem 9085 and gives the user some valuable information in 3 different orientations. The paper could be published in the Materials after some minor corrections

  1. section 3.1 Experiment setup;   the reference # is missing in raw 148.
  2. Table 3 and in along the paper- the data in the table is difficult to follow to the reader the table should include the applicable time, the given print direction, and the annealing temperatures and not in index parameters. 
  3. Figure 2 and Table 4 – l2 is less important in the tensile test as indicated in ISO 527-2, measuring the elongation is correlate with the gauge length l0 and the length of the parallel side portion l1. These values should be added to the measured data.
  4. Section 4.1 –(In addition to remark 3 – what was the lengths used during tests li
  5. Table 8 – see my remark for table 3
  6. Figure 7 - the legend and symbols are similar to IM and AM !!
  7. Figures 8 and 9 – there are some differences in the presented symbols, however, it is hard to see this difference and should be improved.

Author Response

Refer to the attached rebuttal letter.

Reviewer 4 Report

The paper provides a study of low-temperature thermal annealing conducted on ULTEM 9085 fused filament fabrication coupons.

The study was carried out by employing a modified orthogonal array design, the coupons were annealed with a specialized support structure which was co-printed during the manufacturing process.

The evaluation of the mechanical properties of the produced specimens highlighted improvements in tensile strength and reduction in strain concentration. The specimens produced in the present study are characterized by enhanced properties, making the employed annealing methodology a valuable technique to produce FFF parts without significant distortion. The obtained results make the present manuscript a notable contribution to the current literature concerning the produced materials' additive manufacturing processes and performances.

A revision before publishing is recommended according to the following points.

  • - General comment

The authors should improve the literature review by citing other research papers presented in international journals dealing with similar topics and extend the introduction with recent and relevant papers. See for example:

  • Role of infill parameters on the mechanical performance and weight reduction of PEI Ultem processed by FFF. Materials & Design, 193, 108810.

  • Multi-material additive-manufacturing of sustainable innovative materials and structures. Polymers, 11(1), 62.

- Minor points

  • Please ensure to center all figures and tables with all the related captions regarding all tables and figures. Also, do not forget to center the formulas (formulas 1 and 3, for instance, do not seem correctly centered).

  • Figure 5 could be represented better. A good idea could be resizing the picture containing the X, Y, and Z-axis so that the total height of the figure would be as high as the picture (f).

  • Please try to enlarge slightly the graphs contained in figure 6. You could try to use the same graphs disposition used for figure 10 for the first 2 graphs, reporting the third graph just below the other two in a centered position.

  • The text from line 378 to line 383 does not have the same formatting and text size as the main manuscript text.

  • All references should be numbered consecutively and citations of references in text should be identified using numbers in square brackets (e.g., “as discussed by Smith [9]”; “as discussed elsewhere [9, 10]”). Your text already meets this requirement, but the numbers are only written in the main manuscript text, while the references at the end of the document are not numbered. Please assign the correct numbering to the references list.

  • Please check the conformity of the document with the guidelines of the journal (authors’ names or references for example)

Author Response

Refer to the attached rebuttal letter.

Round 2

Reviewer 1 Report

The paper in the revised form can be published, because it fulfills all the conditions of a valuable scientific paper.

Author Response

Thank you for the positive feedback.

Reviewer 2 Report

Nothing to add

Author Response

Thank you for the positive feedback.